# Affective Responses to Barbell-Based Resistance Training in a 16-Week Barbell-Based Strength Training Program for Recreationally Active Adults

**DOI:** 10.3390/sports13030088

**Published:** 2025-03-14

**Authors:** Vanessa M. Martinez Kercher, Janelle M. Goss, Janette M. Watkins, Liam A. Phillips, Brad A. Roy, James F. Dow, Lilian Golzarri-Arroyo, Kyle A. Kercher

**Affiliations:** 1Department of Health & Wellness Design, School of Public Health-Bloomington, Indiana University, Bloomington, IN 47405, USA; vkercher@iu.edu; 2Department of Kinesiology, School of Public Health-Bloomington, Indiana University, Bloomington, IN 47405, USA; jangoss@iu.edu (J.M.G.); janhynes@iu.edu (J.M.W.); 3Program in Neuroscience, College of Arts and Sciences, Indiana University, Bloomington, IN 47405, USA; 4United States Military Academy, Westpoint, NY 10996, USA; liam.phillips@westpoint.edu; 5Logan Health Medical Center, Kalispell, MT 59901, USA; broy@logan.org; 6Biostatistics Consulting Center, School of Public Health-Bloomington, Indiana University, Bloomington, IN 47405, USA; jfdow@iu.edu (J.F.D.); lgolzarr@iu.edu (L.G.-A.)

**Keywords:** strength training, affective responses, barbell training

## Abstract

Background: Despite the well-established physical benefits of resistance training (RT), only 31% of U.S. adults meet RT guidelines, with women participating at lower rates. While the physiological aspects of RT are well researched, less is known about the psychological factors, such as affective responses (e.g., enjoyment, energy). This study explored the relationships between self-efficacy, self-determined motivation, affective responses, and adherence in a 16-week barbell-based RT program. Methods: A prospective longitudinal study was conducted with 43 adults (M age = 45.09 ± 10.7, 81.8% female) enrolled in a community-based RT program. Affective responses were measured pre- and post-training, within RT sessions, and over time. Repeated-measures ANOVA and correlational analyses were used to examine relationships between psychological variables, affective responses, and adherence to the program. Results: Participants reported significant improvements in positive affective responses post-training and across the program’s duration. Self-efficacy and intrinsic motivation were positively associated with higher affective responses and greater adherence. Strength exercises elicited more positive affects compared to power exercises, and lifting heavier relative loads was correlated with more favorable emotional outcomes. Conclusions: The study highlights the importance of psychological factors, such as self-efficacy and motivation, in the relationship of affective responses to RT.

## 1. Introduction

Resistance training (RT) is associated with a decrease in mortality and improvement in physical function [1,2]. Despite the well-acknowledged physical benefits of RT (e.g., increased muscular hypertrophy, strength, endurance, endorphins), only 31% of adults in the United States follow the Center for Disease Control and Prevention guidelines and recommended strategies on muscle strengthening physical activity [3,4]. Women, in particular, are less likely to participate in RT (e.g., 20% meet RT guidelines) due in part to perceived barriers of required time and effort [5,6]. Thus, this general lack of participation in RT begs the question: why are so few people participating in RT on a regular basis? Beyond physiology, part of the answer may lie in our limited understanding of and emphasis on psychological responses to RT.

Affective responses, defined as the emotion experienced in response to a stimuli or situation [7], are important for initiating and maintaining RT, yet the majority of strength and conditioning programs focus entirely on physiological variables like frequency, intensity, load, and volume, without reporting on or equally emphasizing affective responses like enjoyment, pleasure/displeasure, or energy/tiredness (i.e., mood) [2,6,8]. While the relative emphasis on exercise physiology is greater than the emphasis on exercise psychology, the importance of affective responses has been acknowledged in research [9,10,11]. For example, higher levels of affective responses were correlated with greater levels of physical activity and achieving physical activity goals in women [12,13]. Current literature also supports that when exercise produces positive affective trends people are more likely to be consistent with their exercise routine [8,9,14]. Some of the affective responses from a well implemented exercise program include enjoyment, mood, confidence, and self-efficacy [11,12,13], but less studies have focused on the longitudinal relationship between RT and affective responses in periodized RT programs (e.g., strategic variation of intensity, volume, and exercise selection).

Self-efficacy is a potentially important construct for gaining a greater understanding of affective responses to RT. Research supports the concept that self-efficacy (i.e., one’s confidence to perform an exercise) impacts immediate affective responses [15]. However, there are limited data on affective responses within RT sessions, particularly with compound barbell exercises. Additionally, there is limited understanding of the longitudinal association between RT affective states and adherence to RT [16]. Despite our limited knowledge of affective responses to RT, research indicates that immediate affective states (i.e., feelings during exercise) have a greater impact on adherence compared to post-exercise affective states [16]. However, the relationship between acute affective states and adherence requires further investigation. There are various gaps in our understanding of the relationship between affective states and physical activity. For example, few studies specifically focus on RT. Studies that do focus on RT often use single joint movements (e.g., bicep curl) or stationary machines (e.g., leg extension, chest press) [17]. These studies also showcase limited participant background information (i.e., previous experience that could impact affect states during exercise).

In addition to self-efficacy, self-determined motivation may play a critical role in affective responses to RT. Self-determination theory provides a model for understanding how motivation influences human behavior. Behavioral regulation is a concept within self-determination theory that describes the type of self-determined motivation ranging from amotivation to the most intrinsic form of motivation (i.e., relative autonomy) [18]. While physical activity in general (i.e., aerobic activities) has been shown to positively impact an individual’s psychological state [19], data are limited in explaining the relationship between behavioral regulation and barbell-based RT. Research supports a positive correlation between individuals feeling in control of their actions during physical activity and self-efficacious affective responses and exercise adherence [20]. When individuals are intrinsically motivated, they tend to have greater well-being which corresponds with adherence to physical activity. Different forms of physical activity may impact affective states more positively or negatively. For example, functional RT (i.e., free weights) was shown to elicit greater positive affective states compared to machine-based RT [13]. There is a potential opportunity to impact non-competitive adults by increasing our understanding of the relationship between RT and the immediate affective responses.

To build on the body of knowledge around psychological responses to barbell-based RT, we conducted a 16-week prospective longitudinal study of adults enrolled in an existing community-based RT program called Competitive Edge. Competitive Edge was a barbell-based RT program conducted within an 8000+ member medical fitness center in the Pacific Northwest. The program focused on enhancing muscular strength and power by incorporating barbell-based RT. Strength and conditioning professionals certified by the National Strength and Conditioning Association (NSCA), American College of Sports Medicine, and/or USA Weightlifting developed the programming. This study was made possible by a collaboration between the principal investigator, executive director of the medical fitness center, and Competitive Edge supervisor.

This study examined the relationships between (a) self-efficacy, (b) self-determined motivation, (c) acute affective responses, (d) intention to continue RT, and (e) RT adherence during a 16-week RT program. The primary objective of this study was to assess affective responses to RT (a) before-and-after training sessions, (b) within RT sessions, (c) and across time during the 16-week training program. Our primary hypothesis was that participants would experience positive affective responses during and following barbell-based RT sessions. The secondary objective of this study was to investigate the relationship between relative load (weight lifted versus participant body weight), RT self-efficacy, and affective responses. Our secondary hypotheses were that (1) those who lifted more weight relative to their body weight, and (2) those with greater RT self-efficacy would demonstrate more positive affective responses. Our third objective was to determine whether RT self-efficacy and intrinsic motivation (i.e., enjoyment) were related to greater intention to continue RT. An exploratory objective was to identify the exercises that were related to the most positive affective responses. Our hypotheses were that participants with greater RT self-efficacy and intrinsic motivation (i.e., enjoyment) would have greater intention to continue RT. Overall, this study aimed to evaluate immediate affective responses that could influence an individual’s engagement in RT, thereby promoting longevity in RT. To our knowledge, this is the first study integrating self-determination theory and self-efficacy to assess immediate affective responses within RT sessions, pre-post session, and pre-post program in everyday adults over a 16-week longitudinal period in everyday adults.

## 2. Materials and Methods

### 2.1. Sample and Setting

A convenience sample of 43 adults (M age = 45.09 ± 10.7) were enrolled in this quasi-experimental longitudinal study. The sample included 81.8% females (*n* = 35) and 18.2% males (*n* = 8). This longitudinal observational cohort study without a control group took place in a medical fitness center located in the Pacific Northwest in the United States. The medical fitness center had over 8000 members and offered personal training, physical therapy, clinical wellness coaching, group performance training, group exercise classes, cardiopulmonary rehabilitation, swimming, and human performance laboratory services. The study was conducted in a semi-private turf space equipped with open flooring, squat racks, barbells, free weights, pull-up bars, and other standard functional training equipment. Participants were asked to attend three RT sessions a week and were encouraged to attend no less than two training sessions per week. Training groups consisted of 4–9 participants and were scheduled to meet regularly either Mondays and Wednesdays or Tuesdays and Thursdays. A third session per week was offered on Friday and Saturday mornings. RT sessions lasted approximately one-hour.

Inclusion criteria were: (1) members of the medical fitness center; (2) males or females of any race or ethnicity who are at least 18 years old; (3) cognitively able to participate in all research elements as determined by pre-research questionnaire; and (4) free from medical limitations as indicated by initial health questionnaires. Exclusion criteria were as follows: (1) medical conditions that are not appropriate for RT activities; (2) exercise limitations that would affect the participants ability to perform RT activities; (3) acute, uncontrolled cardiovascular, metabolic, respiratory, or neurological conditions; (4) history of stroke in the past 12 months; (5) wheelchair bound, need physical assistance with ambulation or have weight-bearing restrictions; and (6) women that are or will be over 16 weeks pregnant.

### 2.2. Training Program

The RT program included supervised exercise, self-selected loads, and a small group environment (e.g., 4–9 participants in each class). The classes accommodated participants of all skill levels and were structured in two 8-week cycles (16 weeks in total) with clients attending 2–3 classes per week. Within each 8-week session there were two 4-week phases, each with an emphasis on either muscular hypertrophy or muscular strength development. Coaches changed phases every four weeks in line with NSCA-recommended phase principles and to avoid the ‘honeymoon’ effect, which refers to a temporary boost in performance or motivation when a new program is introduced, often followed by stagnation as the novelty wears off [21].

Workouts incorporated a 5–10-min warm-up including dynamic exercises were varied at each intervention phase. Between each of the 8-week sessions, there was a rest week where clients were encouraged to perform active recovery (physical activity outside of their normal RT routine).

### 2.3. Design

The study assessed a 16-week RT program. Week 0 baseline assessments were followed by eight weeks of RT, one week of recommended active recovery, and eight more weeks of training. For simplicity, we call the first eight weeks of training week one through eight and the second eight weeks of training week nine through sixteen. Maximal strength testing was conducted at baseline and 16 weeks. Affective responses (described below in Measures section) were assessed before and after training sessions during even numbered weeks (i.e., weeks 2, 4, 6, etc.) and after each set of two primary compound barbell exercises (see Table 1) during even numbered weeks. Loads and perceived exertion were collected during all even-numbered weeks for the two primary compound barbell exercises. The independent variables for this study were type of lift (strength or power), load lifted, rating of perceived exertion (RPE), behavioral regulation, and self-efficacy. The dependent variables were multiple affective responses (described below), and intention to continue RT. See Figure 1 for Study Design Flow and Table 2 for participant characteristics.

### 2.4. Measures

#### 2.4.1. Strength Testing

Strength testing occurred pre- and post-program and included 3–5 repetition maximal efforts for the deadlift, front squat, and bench press, as well as maximum repetitions for the push-up and pull-up.

#### 2.4.2. Self-Efficacy

Self-efficacy was assessed pre- and post-program using an overall Resistance Training Self-Efficacy (RT-SE) scale. The RT-SE scale was developed and modified by our team in line with Bandura’s approach to perceptions of self-efficacy using 13 items assessing participants’ beliefs on three subscales: mastery experiences (4 items), physical capability (4 items), and resilience (5 items) [22]. Participants were asked to rate their degree of confidence in their ability to perform exercises from 0% (cannot do at all) to 100% (highly certain can do) [22]. The items within the scale were modified to focus on resistance/strength training efforts and congruent with the guidelines suggested by Bandura [22]. The internal consistency and reliability of the measures have been provided among exercise [23,24] and sport-specific [25,26] populations.

#### 2.4.3. Motivation Regulation

The Behavioral Regulation in Exercise Questionnaire-3 (BREQ-3) was used pre- and post-program to assess Deci and Ryan’s (1985) conceptualization of the motivation continuum in line with self-determination theory [27]. The BREQ-3 measures external, introjected, identified, integrated, and intrinsic motivation using a 5-point Likert scale ranging from 0 (not true for me) to 4 (very true for me). The BREQ-3 has demonstrated adequate reliability and validity in adult exercisers [28]. In line with our previously published work, we used a single cumulative score by summing subscale scores to assess how self-determined participants felt [29].

#### 2.4.4. Affective Measures

Affective responses were assessed by validated psychometric scales, including the Feelings Scale (FS), Felt Arousal Scale (FAS), the Activation Deactivation Adjective Checklist (AD ACL), and Physical Activity Enjoyment Scale (PACES) [12,30,31]. These measures were collected during weeks 2, 4, 6, 8, 10, 12, 14, and 16 of the training program and are each described below.

The Feeling Scale (FS) [32] and the Felt Arousal Scale (FAS) [33] were used to assess affective states [8]. The FS is a bilateral scale with an 11-point, single-item assessing pleasure/displeasure with response options from ‘Very Good’ (+5) to ‘Very Bad’ (−5). The FS has demonstrated adequate reliability and validity for assessing affective valence [32].

The Felt Arousal Scale (FAS) [34] assessed low to high arousal on a 6-point scale. The FAS is cited routinely in literature to assess perceived activation in various forms of physical activity. Similar to the FS, the FAS has also been used extensively in exercise settings and has demonstrated adequate reliability and validity [35].

The Activation Deactivation Adjective Check List (AD ACL) [36] assesses affect using 20-items. There are four subscales (i.e., tension, calmness, energy, tiredness) and each have five items rated on a 4-point scale ranging from definitely do not feel to definitely feel. [36]. The AD ACL is a satisfactory option for assessing affective responses to physical activity [8].

The Physical Activity Enjoyment Scale (PACES) [37,38] assessed overall enjoyment with physical activities on 18 bilateral items using a 7-point scale. Examples of response options include “I enjoy it (1) … I hate it (7)” and “I dislike it (1) …. I like it (7)”. Possible scores range between 18 and 126.

Rating of Perceived Exertion (RPE) was assessed with The Borg’s Scale [39]. This scale uses a 15-point Likert scale from 6 (“minimal effort”) to 20 (“maximum effort”). Participants identify their perception of how hard exercises feel. We refer to affective responses and perceived exertion as psychophysiological responses.

Intention to continue RT was measured using two questions, one for intention to continue RT 2 days per week and the other for intention to continue RT 3 days per week. These frequencies of continuing RT were selected because the program goal was for participants to continue training at least 2 or 3 days per week. Each question was assessed on a Likert scale of 1 to 7 with responses ranging from ‘Very Unlikely’ (1) to ‘Very Likely’ (7). The item stated the following: Rate how likely you are to train during the next month.

### 2.5. Procedure

Participants came to the facility on two separate occasions to collect measures before (Week 0) and after the 16-week barbell program. At baseline a written informed consent was obtained in accordance with Institutional Review Board Policy, after researchers provided a written and verbal explanation of the study. Once informed consent was provided, participants were asked to complete two questionnaires, the PAR-Q and a health history to identify contradictions to exercise.

Questionnaires were collected from the participants on the first and last visit. Following collection of psychological measures, participant anthropometrics and body composition were measured. During the first visit, participants underwent a familiarization session to teach them to use their individualized workout sheet and where to report important psychophysiological measures and record load (weight) lifted. The participants received detailed instructions for completing psychophysiological responses (i.e., RPE, FS, FAS, and AD ACL). Additionally, visual prompts were posted along the walls to remind participants of the appropriate scales to utilize during their RT sessions. Participants went through familiarization training on how and where to report their psychophysiological responses.

During weeks 1–16 of training, we assessed psychological responses pre- and post-workout using the Activation-Deactivation Adjective Check List (AD ACL). During weeks 2, 4, 6, 8, 10, 12, 14, and 16, in addition to the pre-post ADACL, participants reported their psychological responses on the Feeling Scale (FS), Feeling Arousal Scale (FAS), and RPE (Borg Scale) after executing each set of their two primary barbell exercises in each week. These primary RT exercises varied by each day within each four-week training phase and included the complex/compound, multi-joint barbell exercises. The exercises were typically variations of Olympic lifts. They included a variety of different barbell squats, deadlifts, cleans, pressing, and pulling exercises (see Table 1).

Familiarization weeks included weeks 1, 5, 9, and 13 (always the first week of a new training phase). During the familiarization weeks we did not assess the within-session measures (FS, FAS, RPE) but they continued to report on the AD ACL pre- and post-RT session. Although these were familiarization weeks, the participants still completed the entire training session each day. We chose not to assess in-class measures because participants would be learning a new routine and exercises that they may or may not have executed before.

Participants were responsible for self-selecting their loads during lifts but coaches supervised the process. Coaches were encouraged to allow the participants to self-select, but coaches had the ability and rapport to influence the weight lifted. This guided self-selected load approach was used to enhance a sense of autonomy within the participant while using the coaches’ judgment and experience in the process.

### 2.6. Data Analysis

Descriptive statistics, (i.e., means, standard deviations, and bivariate correlations) were calculated to provide a descriptive profile of the sample (mean ± SD). To compare the change (post-/pre-session) on affective responses, a linear mixed model was performed with time as fixed effect and subject as a random effect to account for repeated measures. Within session analysis was performed with a linear mixed model using the mean scores over sets. Later, we adjusted for set type (i.e., power or strength-based exercise) to explore if differences on the change score existed based on whether exercises were power- (e.g., hang clean) or strength-based (e.g., deadlift). Pearson correlations were calculated to explore the pairwise relationship between self-efficacy with FS and FAS pre-session. The level of significance was *p* < 0.05. Data were analyzed using R version 4.4.0 (R Core Team 2024) using RStudio (RStudio 2024.04.2).

## 3. Results

### 3.1. Descriptive Statistics

Table 2 shows the participant characteristics for the final sample of adults. The sample was 98% White and 100% non-Hispanic. The post-program body mass index for men (*n* = 8) was 24.8 ± 1.2 lbs and for women (*n* = 35) was 26.8 ± 4.6 lbs.

### 3.2. Objective 1: Affective Responses to RT

When comparing the mean of the pre- and post-sessions, FAS and Energy (measured with the ADACL) significantly increased (Table 3), while Calmness marginally significantly decreased from pre- to post-session. When comparing the change (post-/pre-sessions) between affective responses over time, there were no time effects for FS (*p* = 0.474), while FAS had a significant change across sessions (*p* = 0.005) where FAS increased over time.

For within session, the means of FS and FAS were calculated across sets and compared over time accounting for the set type. FS scores changed both across sessions and set type (*p* < 0.001) with FS significantly increasing over time with more sessions. When analyzing FAS within session, it showed no change across sessions (*p* = 0.592), but did change by set type (*p* < 0.001) (Figure 2).

### 3.3. Objective 2: Relative Load, Self-Efficacy, and Affective Responses

When calculating the correlations for self-efficacy scores (i.e., Mastery, Resilience, Physical) with FS and FAS, there were not significant correlations with the pre-session values. However, self-efficacy scores were significantly correlated with FS and FAS post-session (Figure 3).

The figure below shows the correlation between load lifted and FS, FAS, where there is a significant positive correlation, where the greater the load lifted, the greater the scores of FS (r = 0.20; *p* < 0.001) and FAS (r = 0.15, *p* < 0.001) (Figure 4).

### 3.4. Objective 3: Self-Efficacy, Motivation Regulation, and Intention to Continue RT

For our third objective, none of the three self-efficacy subscales were significantly correlated with intention to continue RT (Figure 5). For motivation regulation, identified (0.21; *p* = 0.060) and intrinsic (0.33; *p* = 0.003) motivation subscales were significantly correlated to intention to continue RT 3 days per week (see Figure 6).

### 3.5. Exploratory Objective: Affective Responses to Different RT Exercises

Our exploratory objective descriptively identified the deadlift as the RT exercise with the most positive affective scores for the FS (3.58 ± 1.25) and FAS (4.71 ± 1.13). In general, many of the strength exercises (less explosive) scored higher on the FS and FAS than the power exercises (more explosive) (See Table 4 for details).

## 4. Discussion

The present study investigated the relationship between affective responses in everyday adults (a) between RT sets, (b) pre-post RT sessions, and (c) over the course of a 16-week barbell-based RT program. This study revealed four key findings: (1) participants felt more energized and experienced more positive affect following the training sessions, as indicated by an increase in FS and FAS scores from pre- to post-session; (2) positive affective responses increased over time within sessions; (3) affective responses were higher during strength exercises compared to power exercises; (4) a positive correlation was found between self-efficacy and affective responses post-training program, but not before the program. Collectively, these findings provide insight into the complex interplay between psychological factors and RT experiences, offering valuable insights to guide future studies with more controlled research designs and generalizable samples.

The first finding revealed changes in affective responses from pre- to post-session, as evidenced by an increase in FAS and FS scores. Essentially, participants felt more energized and had more positive emotions following the training session. This is a similar phenomenon that has been extensively cited in literature regarding aerobic training. For example, running is widely recognized for its psychological benefits, including enhancements in mood, increased feelings of happiness, and a more positive outlook. Research indicates that running not only elevates mood and overall well-being but also alleviates negative emotions such as anger, depression, and aggression [40]. Similarly, some studies show that RT can increase positive emotions and promote a higher sense of well-being [41], although these studies are less frequent compared to those focusing on aerobic exercise. Additionally, our findings support previous findings that using FS and FAS in combination may provide a more comprehensive understanding of the emotional landscape during an RT session [34]. In summary, RT was significantly correlated to more positive affective states, and the FAS and FS scales are synergistic tools for assessing and understanding these positive emotional changes.

The second finding was that within-session FS scores increased over time with more sessions. As weeks progressed and participants experience in RT increased, their FS scores increased, suggesting that gaining experience may contribute to more significant improvements in affective states. While previous research has established that RT can help alleviate symptoms of depression and anxiety [42], the impact of individual experience levels, such as whether someone is a beginner or an experienced weightlifter, is less understood. Current studies suggest that prior experience in RT does not necessarily correlate with affective states [43]. However, our findings indicate that experience may impact the extent of affect changes experienced during RT. These findings suggest that increased experience in resistance training may enhance positive affect, which could support long-term participation by fostering more enjoyable workout experiences over time.

The third finding was that FAS and FS scores improved from pre- to post-sessions, with higher scores observed for strength exercises compared to power exercises. This difference may be explained by the load lifted, as greater loads were associated with higher FAS and FS scores. For instance, power movements require more skill and proficiency to lift weights efficiently, while strength exercises like the deadlift involve lifting heavier loads regardless of the lifter’s experience. The data support this explanation, with exercises such as the deadlift, hex bar deadlift, and back squat showing the highest FS and FAS mean scores. Although existing literature indicates that RT improves affective states, specific variables such as load and volume have not been analyzed separately [42]. Our findings suggest that increasing the load in RT may enhance changes in affective states. Additionally, our results challenge some studies that found anxiety reduction in RT occurs when lifting 70% or less of an individual’s one-rep max [44]. However, our results align with other research indicating that heavier loads can positively impact mood more than lighter loads [45]. Further research is needed to understand how different RT variables impact affective states. Ultimately, our data showed that FS scores vary by RT exercise type (i.e., power versus strength) and over time, while FAS scores vary by RT exercise type but remained consistent over time.

The fourth finding was that self-efficacy did not correlate with core emotions of pleasure/displeasure (i.e., FS) and levels of activation (i.e., FAS) pre-program, but there was significant correlation post-program. The current literature on FS and FAS in RT contexts remains scarce in comparison to literature on psychological responses to aerobic training. One possible explanation for this could be that pre-program, individual stressors impacted FS and FAS and potentially self-efficacy, whereas post-program participants had gone through a similar stimulus (i.e., the training program) and it regulated some of their affective responses. Most studies report improvements in self-efficacy from pre- to post-exercise programs, but this may reflect publication bias, as non-significant findings are less likely to be published [46]. Therefore, the point here is not about self-efficacy improving pre-to-post, but rather the absence of a significant correlation pre-program and a strong correlation post-program. We argue that part of the benefit of an RT program may be that it helps participants gain psychological congruence between self-efficacy and affective states (i.e., pleasure/displeasure, activation). Many past studies have assessed the relationship between self-efficacy and exercise-related outcomes [47], but less have reported on the changes in the relationship between self-efficacy and affective states pre- to post-program. Taken together, our finding of the strong correlation of self-efficacy to pleasure/displeasure and activation post-program extends the current literature by highlighting a potentially positive affective relationship in primarily White and relatively affluent recreationally active adults that participate in a barbell-based RT program.

This study had multiple limitations. First, internal validity is limited without a randomized controlled design. The lack of a control group and selection bias (e.g., convenience sampling) limits the interpretation of the study’s results. The findings must be considered as correlational rather than causational as well as interpreted conservatively. The main reason the convenience sampling method was selected was because this was a preliminary evaluation of a community-based program. Future studies should aim for research designs with greater internal validity. Second, there is a lack of generalizability in the sample because of the primarily White sample and relatively affluent participants that had to be able to pay for program and facility membership to participate in the study. Future studies should aim to replicate these findings in other populations as we recognize the potential for individual and cultural differences to impact results. Third, as this study used self-report measures, the presence of response bias is likely. Fourth, it is possible that other confounding factors (e.g., previous RT experience) could have influenced the results, had they been included in the data collection and analysis. Future studies should consider including other potentially important confounding variables, such as previous RT experience. Despite these limitations, the present study offers novel findings through access to a unique population of primarily White and relatively affluent recreationally active adults, within-session measures, a longitudinal design, and primarily complex barbell-based exercises rather than single joint or commonly assessed RT (e.g., leg extension, bicep curl, bench press, leg press). Researchers examining the relatively unexplored area of psychological responses to barbell-based RT in other populations should consider these findings while aiming to employ research designs including more generalizable samples and overcoming methodological limitations.

## Figures and Tables

**Figure 1 sports-13-00088-f001:**
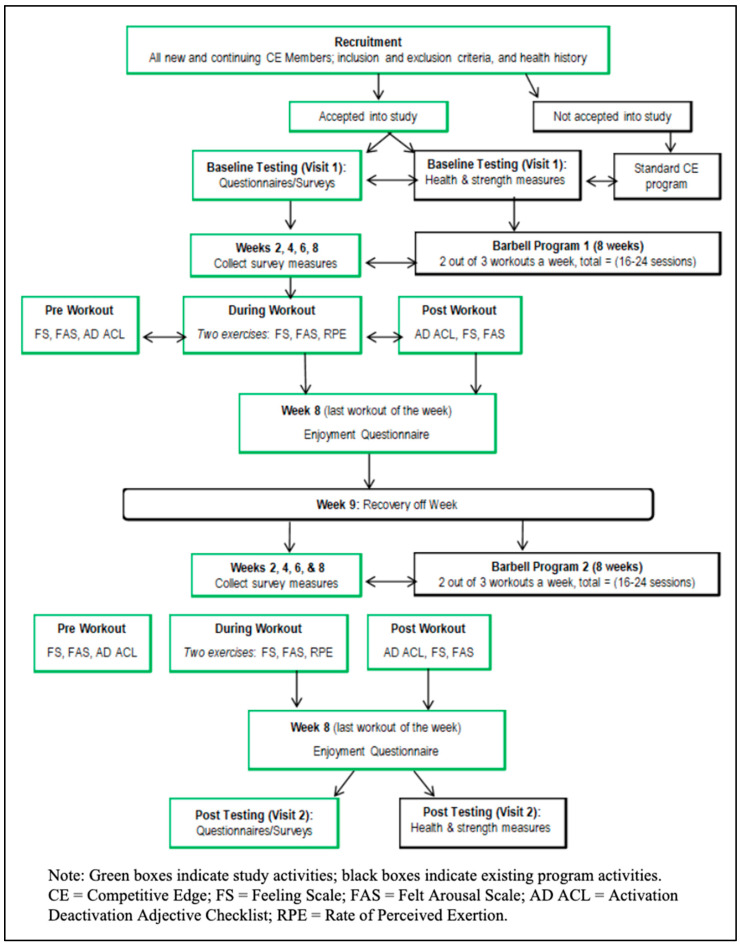
Study design flow.

**Figure 2 sports-13-00088-f002:**
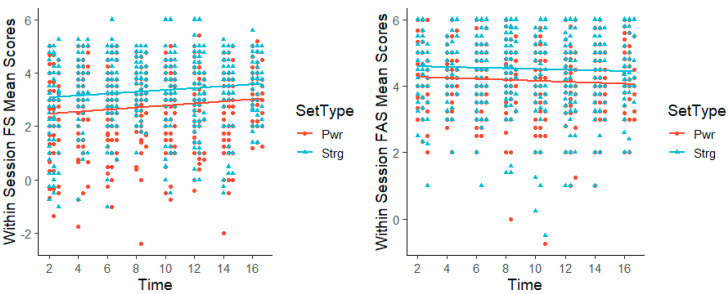
FS and FAS mean scores over time by set type.

**Figure 3 sports-13-00088-f003:**
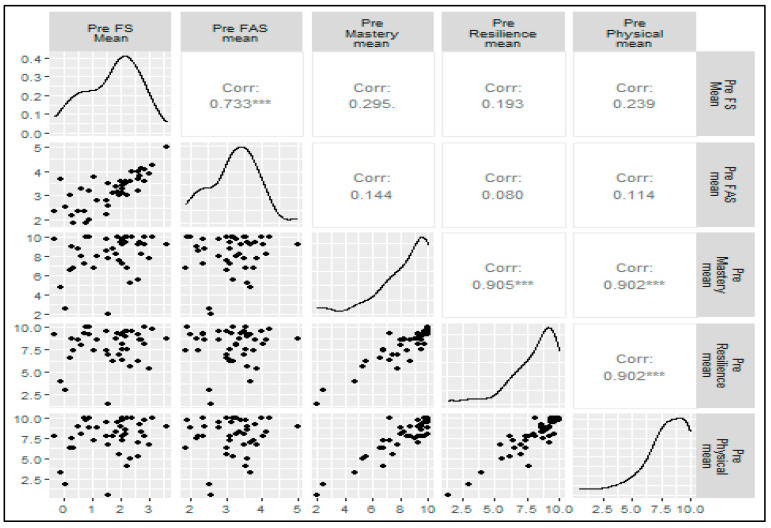
Correlation for self-efficacy scores versus FS and FAS. Note: *** = *p*-value < 0.001; ** = *p*-value < 0.01; * = *p*-value < 0.05; . = *p*-value < 0.10.

**Figure 4 sports-13-00088-f004:**
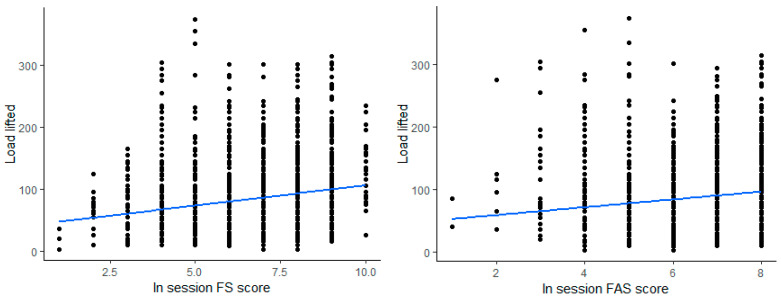
Correlation for load lifted versus FS and FAS.

**Figure 5 sports-13-00088-f005:**
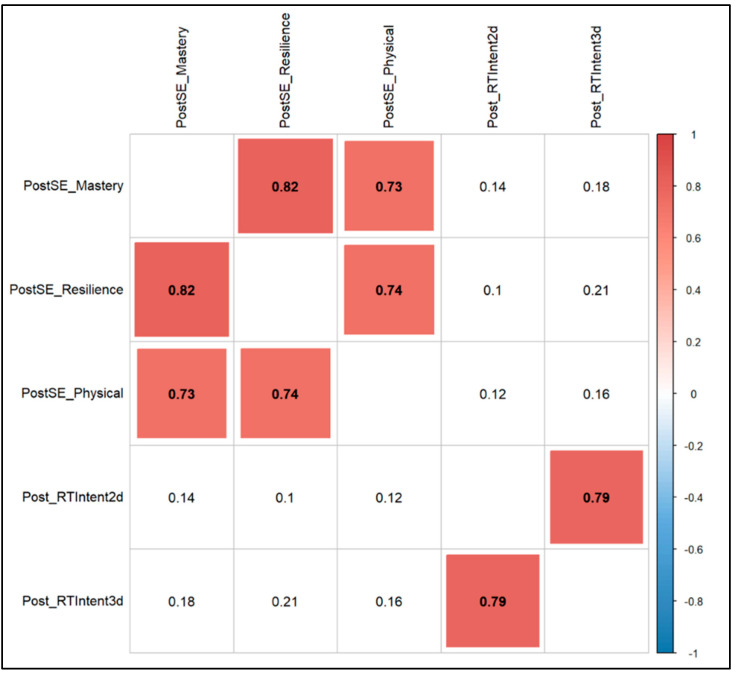
Correlations between self-efficacy and intention to continue RT.

**Figure 6 sports-13-00088-f006:**
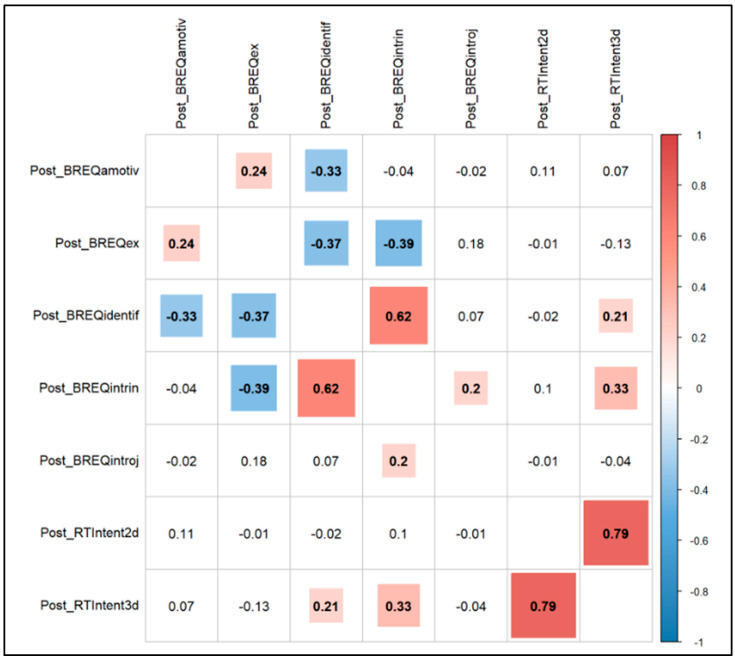
Correlations between motivation regulation and intention to continue RT.

**Table 1 sports-13-00088-t001:** Complex barbell power and strength exercises.

Complex Power	Complex Strength
Hang clean *	Deadlift *
Power clean	Front squat *
Hang high pull	Bench press *
Push press	Back squat
Snatch	Split squat
Split jerk	Hex bar deadlift

Note. * Indicates a movement included in the 3–5 rep max testing.

**Table 2 sports-13-00088-t002:** Participant characteristics.

Variable	Male (*n* = 8)	Female (*n* = 35)
Age	43.6 ± 10.5	45.4 ± 10.9
Education (%)		
Some high school	0%	2.8%
Trade/vocational certification	12.5%	0%
Some college credit	25%	2.8%
Associate degree	0%	8.5%
Bachelor’s degree	25%	42.8%
Some graduate school	0	2.8%
Master’s degree	37.5%	34.2%
Doctorate degree	0%	5.7%

**Table 3 sports-13-00088-t003:** Pre-/post- FS, FAS, ADACL results.

Metric	Pre-Session Mean (SD)	Post-Session Mean (SD)	*p*-Value
FS	1.68 (1.48)	3.66 (1.29)	0.839
FAS	3.17 (1.04)	4.95 (0.96)	<0.001 *
Energy	2.01 (0.76)	3.23 (0.69)	0.008 *
Tired	2.38 (0.91)	1.52 (0.59)	0.523
Tension	1.62 (0.62)	1.73 (0.57)	0.293
Calmness	2.10 (0.78)	1.85 (0.71)	0.063

Note: * indicates significant *p*-value.

**Table 4 sports-13-00088-t004:** Mean affective response scores after sets of various RT exercises.

Exercise	FS	FAS
Deadlift	3.58 (1.25)	4.71 (1.13)
Back squat	3.52 (1.19)	4.49 (1.15)
Front squat	3.32 (1.40)	4.22 (1.17)
2 count pause bench press	3.28 (1.39)	4.40 (1.03)
Tempo front squat	3.25 (1.42)	4.41 (1.22)
Hang clean *	3.11 (1.28)	4.37 (1.10)
Split jerk *	2.94 (1.52)	4.21 (1.22)
Hex bar deadlift	2.93 (1.45)	4.68 (1.00)
Hang high pull drops *	2.87 (1.45)	4.07 (1.02)
Power clean *	2.83 (1.31)	4.22 (1.08)
BB hang snatch *	2.77 (1.38)	4.07 (0.99)
Power clean to split jerk *	2.52 (1.47)	4.40 (1.10)

Note: Data presented as mean (standard deviation) and sorted in descending order for FS. FS = Feeling Scale; FAS = Felt Arousal Scale; BB = Barbell. * = power exercise.

## Data Availability

The data presented in this study are available on request from the corresponding author due to participant confidentiality and privacy considerations.

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
