# Peer review of "Affective Responses to Barbell-Based Resistance Training in a 16-Week Barbell-Based Strength Training Program for Recreationally Active Adults"

_sports, 2025, doi:10.3390/sports13030088_

Round 1
Reviewer 1 Report
Comments and Suggestions for Authors
Introduction
The authors address a relevant topic in the introduction: affective responses to barbell resistance training in recreationally active adults. They present a clear hypothesis on the relationship between self-efficacy, self-determined motivation, affective responses, and adherence to a 16-week training programme (lines 22–26).
However, there is a lack of justification for the chosen approach (lines 54–60). While the importance of studying affective responses to resistance training is stated, the authors do not argue why this approach is novel or how it differs from previous studies. Likewise, the authors present insufficient or outdated references (lines 35–45), mentioning studies on low participation in resistance training but failing to integrate recent references explaining how psychological factors have previously been addressed in this context.
For this reason, it is necessary for the authors to clearly justify the originality of the study and its contribution to the existing knowledge in the field of resistance training. Additionally, they should include updated references supporting the importance of psychological factors in adherence to resistance training.
Methodology
Regarding the methodology, the authors use a prospective longitudinal design, which allows for the evaluation of the evolution of affective responses over 16 weeks (lines 24–25). Furthermore, the sample includes a diverse range of participants in terms of age and gender, although with a female majority (81.8%), which may be relevant to the study (lines 25–26).
However, there is a lack of randomisation in the sample (line 126), as it is mentioned that convenience sampling was used, which may affect the generalisation of the results. Additionally, there is an absence of a control group (line 128), as no comparison group is included, preventing causal relationships from being established for the observed associations. This is compounded by deficiencies in the validation of instruments (lines 195–198), as the authors do not mention whether the self-efficacy scale used was specifically validated for this type of population.
For these reasons, it is necessary for the authors to justify the limitations of the sampling method and discuss its impact on the applicability of the results (line 126). Likewise, they should consider including a control group in future studies to strengthen the study’s internal validity (line 128). Finally, they should provide information on the validation and reliability of the scales used to measure self-efficacy and motivation (lines 195–198).
Results
The results are presented in detail, showing affective responses before and after training sessions, allowing trends to be observed throughout the programme (lines 303–314). Additionally, the authors use appropriate statistical tests, such as mixed linear models and Pearson correlations, to analyse the relationships between variables (lines 294–300).
However, there may be potential response bias (lines 341–342), as the results rely heavily on self-perception scales, which may introduce biases into participants’ responses. Likewise, there is a lack of confounding variable analysis (lines 359–360), as there is no assessment of potential confounding factors that may influence the relationship between self-efficacy, lifted load, and affective responses. Finally, errors in descriptive tables are observed (line 373), with inconsistencies in some values, which could affect the interpretation of the data.
For these reasons, it is recommended that the authors consider including objective measures to complement self-perception scales and reduce response bias (line 341). They should also conduct confounding analyses to assess whether other variables, such as previous training experience, influence the results (line 359), as well as review and correct errors in the results tables to ensure data accuracy (line 373).
Discussion
In the discussion section, the authors analyse the implications of the findings and relate them to previous studies, which contextualises the relevance of the study (lines 395–406). They acknowledge the influence of self-efficacy and motivation on adherence to resistance training (lines 403–406).
However, some speculative interpretations are made regarding the presented results (lines 407–409), as the authors state that affective responses improve training adherence without considering other variables that may influence this phenomenon. Likewise, there is a lack of discussion on methodological limitations (lines 470–471), as the authors do not address how the lack of a control group and selection bias may have affected the results. Finally, they generalise the results (line 472), assuming that the findings can be applied to any recreationally active population without considering individual or cultural differences.
For this reason, it is recommended that the authors avoid making claims without direct evidence and refine conclusions by considering other possible explanations for the results (line 409). Additionally, they should provide a more detailed discussion of the study’s methodological limitations and their impact on the findings (line 471), as well as exercise greater caution in generalising the results and acknowledge the need for replication in different populations (line 472).
Conclusions
In the conclusions section, the authors clearly summarise the importance of affective responses in resistance training and their relationship with adherence (lines 495–500). They also highlight the relevance of considering psychological factors in the design of resistance training programmes (lines 503–504).
However, they express excessive optimism in their conclusions (line 495), as they suggest that the findings can guide strategies to improve adherence without acknowledging the study’s limitations. Likewise, there is a lack of practical proposals (line 504), as no concrete recommendations are provided for applying the results in training settings.
For this reason, it is recommended that the authors refine their conclusions, acknowledging the limitations of the study and the need for further research (line 495). Additionally, it is suggested to include specific recommendations for coaches and professionals on how to apply these findings in practice (line 504).
Overall Evaluation
The article provides a detailed analysis of affective responses to barbell resistance training, offering valuable insights into their relationship with self-efficacy and motivation. However, its methodology could be strengthened by including a control group, instrument validation, and better bias control. Additionally, the discussion should delve deeper into the study’s limitations and avoid speculative interpretations.
Author Response
Thank you for your thoughtful comments. We have made significant revisions and addressed them point-by-point in the attached document. Thank you again!

Reviewer 2 Report
Comments and Suggestions for Authors
Dear Authors
The work submitted to me for review takes up an interesting topic. It is worth including the issue of the impact of resistance training on the motivation of people exercising in future studies. The work requires minor corrections and additions within individual sections:
Introduction:
-Please add a few sentences in the introduction about the effects of undertaking physical activity, including resistance training, namely the production of endorphins, pleasure as a result of undertaking training. It should be emphasized that wisely implemented physical activity can bring excellent health effects.
Material and methods:
In this section, the very detailed description and procedure of conducting the research deserve recognition, please only complete the information on the criteria used for inclusion and exclusion from the study in this section and describe them
Results:
In tables and graphs, please emphasize statistically significant results
Best regards
Author Response
Thank you for taking the time to review our manuscript. We have uploaded an attachment with point by point responses to your comments.

Round 2
Reviewer 1 Report
Comments and Suggestions for Authors
Dear authors,
having reviewed the new version of your paper, I have seen that you have taken into consideration the recommendations I made to you to continue with the publication process of your research.
New relevant references have been incorporated, the section on the sample has been expanded and some sections, especially Affective Measures, have been rewritten. Likewise, some of the data in the results that could have been inconsistent have been revised and the discussion has been revised in the recommended terms.
It is therefore recommended that the work be revised in its current state.